# Clinical and cost-effectiveness of teen online problem-solving for adolescents who have survived an acquired brain injury in the UK: protocol for a randomised, controlled feasibility study (TOPS-UK)

Jenny Limond,[1] Shari L. Wade,[2] Patricia Jane Vickery,[3] A Jeffery,[4] Fiona C Warren,[5] Annie Hawton,[1] Janet Smithson,[1] Tamsin Ford,[1] Sarah Haworth,[6] Anna-Lynne Ruth Adlam [7]

For numbered affiliations see end of article.

**Correspondence to**
Dr Jenny Limond;
j.limond@exeter.ac.uk

## ABSTRACT

**Introduction** Paediatric acquired brain injury is a leading cause of mortality in children in the UK. Improved treatment during the acute phase has led to increased survival rates, although with life-long morbidity in terms of social and emotional functioning. This is the protocol for a feasibility randomised controlled trial with an embedded qualitative study and feasibility economic evaluation. If feasible, a later definitive trial will test the effectiveness and cost-effectiveness of an online intervention to enhance problem solving ability versus treatment as usual.

**Methods and analysis** Twenty-five adolescents and their families identified by primary or secondary care clinicians at participating UK National Health Service Trusts will be recruited and individually randomised in a 1:1 ratio to receive the online intervention or treatment as usual. Participants will be followed up by online questionnaires 17 weeks after randomisation to capture acceptability of the study and intervention and resource use data. Qualitative interviews will capture participants' and clinicians' experiences of the study.

**Ethics and dissemination** This study has been granted ethical approval by the South West-Exeter Research Ethics Committee (ref 17/SW/0083). Results will be disseminated via peer-reviewed publications and will inform the design of a larger trial.

**Trial registration number** ISRCTN10906069

## INTRODUCTION

In 2012, paediatric acquired brain injury (pABI) was identified as one of the leading causes of death in children aged 5 to 19 years[1]. In the UK, 280 children per 100 000 require at least 24 hours hospitalisation for traumatic brain injury (TBI) each year[2]. When considering other aetiologies such as brain tumours, stroke and infection, there are even greater numbers of children surviving

### Strengths and limitations of this study

► This is the first study to test an online problem-solving tool for adolescents who have survived a brain injury in the UK.
► The study explores the feasibility of recruitment strategies, data collection and economic evaluation to inform the design of a larger randomised controlled trial.
► The effect of the online intervention on executive functioning is not assessed.
► This study is conducted online, with minimal telephone support and no face-to-face contact.
► This feasibility study was conducted over a short time period thus limiting the opportunity for a long-term follow-up. The definitive trial would include a follow-up period of at least 6 months.

pABI.[3] The long-term, even life-long, effects on social functioning, cognition, emotions and behaviour mean that pABI is a leading cause of disability.[4 5]

Despite these on-going difficulties, children with pABI do not automatically receive specialist education, often returning to mainstream schools with little or no additional support. Furthermore, although specialist tertiary National Health Service (NHS) services do exist in the UK, there are limited outreach support or community services for ongoing and emerging difficulties.[6] Families report struggling to access appropriate treatments despite their child developing significant and complex needs. Families also report significant distress and burden when caring for a child who has survived pABI, leading to an increased risk of mental health difficulties

in parents and siblings, and a breakdown in parental relationships.[7] Without appropriate neuropsychological interventions, pABI can lead to increased risk of substance misuse,[8] mental health difficulties,[9] unemployment and criminal behaviour in adulthood.[10] Thus, the long-term costs of pABI to the individual, his/her family and society as a whole can be substantial.[11]

Executive function (EF) difficulties (higher-order cognitive processes that govern goal-directed action and adaptive responses to novel or complex situations) are common following pABI.[12] These difficulties can present later in childhood (particularly in early adolescence), sometimes many years after the initial brain injury. EF difficulties can have far reaching effects, including problems with academic achievement,[13] social communication,[14] emotion and behaviour regulation[15] and peer relationships.[16] Indeed, families caring for children with pABI often report that EF difficulties significantly contribute to their increased levels of stress.[17] Effective interventions targeting child EF and family burden are likely to have significant patient benefit especially as developmental studies show that poor family function can also negatively impact on the child's development of EF abilities.[18]

Despite the clear negative impact of EF difficulties following pABI, there is currently a paucity of research examining the effectiveness of interventions to improve function.[19] In an attempt to address this gap, Wade and colleagues developed an online web-based problem-solving, communication and self-regulation intervention for adolescents (12 to 17 years) who have survived a TBI, and their families (Teen Online Problem-Solving, TOPS).[20] The development of online family problem-solving treatment for paediatric TBI was predicated on evidence of the reciprocity between child recovery and parental psychological well-being and informed by input from survivors of TBI and their families. Problem-solving therapy provides a systematic approach for coping more effectively with life's challenges by creating a positive mindset and developing, implementing and evaluating solutions to problems.[21] Online family problem-solving treatment simplifies the ABCDEs of problem-solving: defining the Aim, Brainstorming, Choosing the best strategy to implement, Doing it (creating a concrete, step-by-step plan) and Evaluating whether it worked. Given the focus on the family rather than the individual, TOPS also focuses on communication skills and collaborative problem-solving. For adolescents with executive function and problem-solving deficits following TBI, the steps of problem-solving also provide an executive function heuristic.[22] Additionally, the programme teaches stress management and metacognitive strategies such as stopping to think before acting and self-monitoring that can serve to promote self-regulation. Similar to the face-to-face, multifamily Brain Injury Family Intervention,[23] online modules provide didactic information regarding the cognitive and behavioural consequences of TBI, training in problem-solving, communication and self-awareness/regulation and exercises to reinforce understanding. After completing the online modules on their own, families meet with a trained therapist to implement to the problem-solving process around an Aim (goal) that they have identified. Thus, the online problem-solving programme was grounded in the scientific literature, tailored to address the unique neurobehavioral and family consequences of TBI and refined through stakeholder input.[20] Problem-solving therapy has also been trialled as a telephone-based intervention for adults with mild TBI and with families of adults with more significant brain injury,[24] suggesting its potential utility across age groups and modes of delivery.[25]

Research to date has been undertaken in the USA, and has focused on children and adolescents who have survived a TBI and their families.[20 26–31] These studies have demonstrated improvements in EF, child behaviour, parental depression and family-child conflict when comparing TOPS with an internet-resource comparison control. Families have reported finding the online delivery of the intervention helpful, making the intervention easy to access at a time that is convenient for them. This is an important consideration when providing interventions for children with pABI in the UK and their families, because brain injury services in the UK often cover a large geographical region (many accepting national referrals), and families often have multiple commitments when caring for these teenagers with complex needs.

Despite this emerging evidence base, it is not yet known whether the gains described can also be demonstrated in adolescents with non-traumatic brain injuries (eg, brain tumour, stroke), when the intervention is delivered remotely (ie, no initial home visit to set up the intervention thus keeping delivery costs to a minimum), and if the intervention is cost-effective relative to treatment as usual (TAU).

A British version of the TOPS programme, (TOPS-UK) has been developed. The didactic material is presented with British English spelling and narration, UK-based information including scenarios, video clips and resource links. TOPS-UK has been further modified to include examples of those with paediatric acquired brain injury as well as traumatic brain injury.

The potential of TOPS-UK to have a positive effect on the lives of thousands of adolescents with pABI needs testing in a rigorous multicentre, randomised controlled trial (RCT) in the UK, allowing a definitive test of its clinical and cost-effectiveness. This feasibility study is designed to determine if a large RCT is feasible, with the ultimate aim of the research being to provide an evidence base for effective neuropsychological interventions to be recommended by the Department of Health guidelines (eg, NICE) for survivors of pABI.

A two-arm randomised controlled feasibility phase II study will be conducted.[32] The aim is to assess whether a larger, fully powered, definitive RCT and cost-effectiveness analysis can be successfully planned and delivered. In order to gain as much information as possible to inform a fully powered RCT, additional objectives are considered

| Table 1 | Study objectives |
|---|---|
| **Number** | **Study objectives** |
| 1 | Are clinicians able and willing to identify participants (eg, has time and access to database, medical records or clinic lists to identify potentially eligible patients)? |
| 2 | Are the online screening, consent and randomisation process feasible? |
| 3 | How many participants at each site are initially eligible, approached and consented, have completed screening and baseline questionnaires, been randomised (with a recruitment target of at least five participants per site), completed treatment and completed outcome assessments? |
| 4 | Do adolescents and parents find the intervention and outcome measures acceptable? |
| 5 | What are parents' and adolescents' experiences of study participation? |
| 6 | Is it possible to calculate means and SD for a full RCT for the potential primary outcome measure? |
| 7 | What resources will be required to run a main trial? |
| 8 | Do some sites need additional support to be able to recruit to target in a future trial? |
| 9 | Do the trial and economic evaluation methods and procedures yield the information required? |
| 10 | How willing are participants to be randomised and complete outcome measures? |
| 11 | How well do participants adhere to the intervention (number of sessions completed)? |
| 12 | How many complete and analysable data sets are yielded, and what is the level of missing data? |
| 13 | How many participants are lost to follow-up? |
| 14 | What is the coach's experience of supporting intervention delivery? |
| 15 | What are adolescent and parent experiences of TAU? (TAU arm) |

RCT, randomised controlled trial; TAU, treatment as usual.

in this feasibility trial, including accessing information related to treatment set-up, delivery costs and resource use (table 1).

## METHODS AND ANALYSIS

This is a randomised, controlled, multicentre feasibility study in young patients with pABI aged 12 to 18 years. Twenty-five participants will be randomised (minimised by site and type of brain injury) in a 1:1 ratio to receive TAU or treatment as usual plus Teen Online Problem-Solving (TAU + TOPS UK). Treatment allocation will not be blinded. The protocol (V.4.1, 30 August 2018) is registered on Current Controlled Trials ISRCTN10906069 and follows Standard Protocol Items: Recommendations for Interventional Trial guidelines.[33]

## Setting

Two settings will be utilised to examine feasibility outcomes associated with recruitment of adolescents known to have had a pABI:

1. Direct referral by clinicians of adolescents who are under the outpatient care (usually neurology, neuropsychology or paediatric) of five participating NHS Trusts in England. Screening logs will be maintained at all sites to record the number of patients screened and enrolled in the study. These sites were selected to be representative of the range of types of services which might be included in a future trial, with TAU ranging from no provision through to services providing multi-disciplinary outpatient care over the long-term.

2. Recruitment from Participant Identification Centres (PICs) in primary or secondary care, or charitable organisations. Potential participants will respond directly to study promotion including leaflets/posters/participant information sheets distributed by parent-led groups, charities or support groups.

Study participation will be supported by research nurses at each participating NHS Trust. For participants directly referred to the study, nurses will make the first approach to the family. For those identified through PICs, families will be asked to contact the research nurse whose details are provided on the patient information sheet (PIS). The research nurse will support all families during the online consent and will be a point of contact for all participants during the study. The research nurse or other research team member will support baseline and follow-up measure completion. The intervention will be provided through online materials and video conference links with a coach trained in the TOPS-UK intervention. The coach will also receive weekly supervision from a qualified clinical psychologist on the research team to ensure treatment fidelity. The single study coach will be centrally based at the lead site.

## Participant inclusion criteria

To be included in the study, participants will be aged 12 to 18 years at the time of recruitment and have survived a pABI. Diagnosis of pABI includes: moderate-to-severe TBI (Glasgow Coma Scale score <12 and/or post-traumatic amnesia >24 hours and/or loss of consciousness >30 min), stroke, brain tumour, central nervous system infection (fungal, bacterial protozoal or viral origin including encephalitis, meningitis, brain abscess, tuberous sclerosis, acute disseminated encephalomyelitis and Guillain-Barre syndrome). Participants must be medically stable (having reached a plateau in recovery following the index injury or illness). They must also have access to the internet, and have EF difficulties in the opinion of the local Principal Investigator (PI). The local PI is not required to undertake any EF assessment for the purposes of the study, but will use their clinical judgement, and infer from any assessments that they have undertaken, whether a participant is likely to be experiencing EF difficulties in their everyday lives. At least one

parent or guardian living with the adolescent must be available for the family to participate. Adolescents with a comorbid diagnosis of attention deficit hyperactivity disorder, autistic spectrum disorder or specific learning difficulties (eg, reading difficulties including dyslexia) will be eligible to be recruited.

### Participant exclusion criteria
Participants will be excluded from the study for the following reasons: insufficient English language, capacity or willingness for the parent/adolescent to consent/ assent to the study; pre-injury or comorbid conditions such as sensory impairments and global developmental delay, known to impair engagement with the computer and treatment materials; or non-accidental brain injury.

### Participant recruitment
The trial complies with the Declaration of Helsinki and Good Clinical Practice (GCP) guidelines. GCP compliance ensures that the rights, safety and well-being of research participants are protected and that research data are reliable. At sites, the local PI, research nurse or other member of the research team in conjunction with the clinical team will identify potential eligible participants, usually from current outpatient clinic lists or review of medical records. An anonymised log of all participants screened with reasons for exclusion will be kept at site. Information sheets for both adolescents and their parents/guardians will be provided to potentially eligible families. The relevant PIS will be given in person to families attending an outpatient clinic, or sent by post with a cover letter from the recruiting clinician. All families will be provided with contact details and informed that a member of the research team will contact them by telephone at least 24 hours after receipt of the information to discuss the study.

During this telephone call, the parent and adolescent will be given the opportunity to ask questions about the study and to confirm whether they are interested in participating. If interested, eligibility criteria will be checked with the parent to confirm suitability for the study and ascertain which parent will participate in the intervention and complete parent-rated outcomes. A summary of what to expect from the online screening and consent process will also be provided.

The process of explaining the study and determining eligibility of potential participants will be undertaken by an appropriately trained member of the research team as delegated by the PI, depending on local arrangements. All staff undertaking this process for this study must have completed GCP training provided by a responsible organisation (eg, University, NHS Trust) and must be authorised by the PI to explain the study and assess eligibility, on the site's study delegation log.

### Online screening and consent
Informed consent and assent will be obtained via the study-specific website, developed and maintained by the

UK Clinical Research Collaboration (UKCRC)-registered Peninsula Clinical Trials Unit (CTU) at Plymouth University.

### Consent/assent
The consent process will depend on the age of the participants:
► Participants aged 16 to 18 years will provide online informed consent.
► Participants aged 12 to 15 years will provide online, informed assent. Those who have given assent and who reach the age of 16 years during the study, will be asked to provide online informed consent.
► Parents of all participants aged 12 to 15 years will provide online informed consent on behalf of their child ('signature' from one parent required).
► All participating parents will provide online informed consent for their own participation in the study.

### Web-based consent process
The research nurse will enter on the study website brief details of those families who wish to participate in the study and who satisfy the initial inclusion/exclusion criteria. This will trigger an email to the adolescent, containing a link to the age-appropriate assent or consent form within the website, which the adolescent should complete.

Following completion of the adolescent assent/ consent form, a separate email will be sent to the nominated parent participant containing a link to the parental consent form. Parents of adolescents aged 12 to 15 years will be asked to consent on behalf of their adolescent in addition to giving consent for their own participation. Two reminder emails will be sent from CTU followed by a 'last chance' email, all with reminders of the study link. If the family has not completed the consent process after 2 weeks, the local research team will be notified and will contact the family to ask if they still wish to participate in the study.

### Face-to-face consent option
If the parent and/or adolescent has not completed the online consent process, CTU will inform the local research nurse who will telephone the family to offer the option of a clinic visit when the nurse can support the parent and/or adolescent to complete online consent/ assent.

### Online BRIEF-2 parent-rating and allocation of study number
Following completion of the consent process, the parent will be prompted to complete the online Behaviour Rating Inventory of Executive Function, Second Edition (BRIEF-2)[34] parent-rating questionnaire. If the parent does not wish to complete this immediately following the consent process, it will be possible to return to the website to complete it later. Two reminder emails will be sent from CTU followed by a 'last chance' email as above. Face-to face support to complete the BRIEF-2 will also be offered by the research nurse, as above. The parent must complete all elements of the BRIEF-2 before progressing

to the baseline measures as this is the proposed primary outcome measure for a definitive trial.

The website will calculate the BRIEF-2 scores. The website will assign a unique four-digit study number to each participant and participants will be identified in all study-related documentation by this study number. An email to both the adolescent and the parent will then be sent, inviting them to follow a web link to complete the baseline measures, and informing them of their study number.

As part of the consent process, adolescents and parents will be reminded via the website that they are free to withdraw from the study at any time without giving a reason and without affecting further treatment.

### Baseline data collection (adolescent and parent self-completion)

On receiving the link to the baseline measures page of the study website, the adolescent and parent will complete their separate baseline questionnaires. The baseline measures should be completed within 2 weeks; if either the parent or adolescent have not undertaken these after 1 week, an email reminder to the parent (in all cases) and the adolescent (if adolescent measures not done) will be sent. In addition to the questionnaires, the parent will provide sociodemographic information, details of the adolescent's past medical history and concurrent medication.

Participants will also be given the option of completing baseline measures with telephone support from a member of the research team. A £15 gift voucher will be sent to each family on completion of their baseline measures.

### Randomisation

Participants will be randomised via a web-based system created by the CTU in conjunction with the trial statistician, using minimisation by study site and type of brain injury (TBI/tumour/other). Participants will be randomly allocated in a 1:1 ratio to receive either TAU or TOPS-UK + TAU. The minimisation process will retain a stochastic element to retain allocation concealment. If any participant is found to be ineligible following randomisation, he/she will be excluded from analyses.

This study is not blinded. Following randomisation, the adolescent, parent, PI and research nurse will be notified by email of the adolescent's treatment allocation, and this information will be held on the study site file. The parent and adolescent will also be advised when to expect further contact from the research team. In addition, the TOPS coach will be notified of all participants allocated to the intervention arm (TOPS-UK + TAU).

### Baseline data collection (research nurse)

The research nurse or another member of the research team at each site will record relevant injury characteristics data for each adolescent (including premorbid/developmental history) in the study-specific web-based case report form.

### Recording study participation in medical notes

The research nurse will make a record of study participation in the adolescent's hospital notes according to local practice, stating that consent was obtained online and face-to-face support was given if appropriate. The research nurse will file a copy of the PIS in the hospital notes along with printed evidence from the study website of the informed consent process and who was involved. The nurse will also send a standard letter to the participant's general practitioner recording study participation.

## INTERVENTION

The study has two treatment arms, 'TAU' and 'TAU + TOPS-UK'. Those allocated to the TAU + TOPS UK arm will work through the intervention programme taking 1 to 2 weeks per module. One coach will be trained to deliver the intervention to all of the participants in the intervention arm. The coach will coordinate the timing of the modules so that all families will have completed the intervention at 16 weeks. This will allow time for other commitments and holidays/breaks to be built into the schedule. The length of time taken to complete each module will be recorded to inform timing of follow-up in the main trial.

### Treatment as usual (control)

TAU will vary at each recruitment site because currently there is no evidence-based treatment for adolescents with pABI and their families. The type of TAU received will be recorded for each participant at baseline via a parent-rated measure of adolescent health and social care resource use (a modified version of the Client Service Receipt Inventory). There will be no opportunity for any TAU participants to receive the TOPS-UK intervention at any point during or after the study.

### Teen online problem-solving (intervention)

TOPS is an online intervention, which is provided by the Cincinnati Children's Hospital website, via a link from the study website. For the purposes of this study, the US-based TOPS treatment content was edited to include British (English) spelling, narration and UK resource links, and modified to suit participants with pABI rather than traumatic brain injury only. All participants allocated to the TAU + TOPS-UK arm will be supported by a single TOPS-UK coach who will make weekly contact by video conference.

For those participants allocated to TAU + TOPS-UK, the TOPS-UK coach will contact the parent and/or adolescent by telephone, introduce themselves, discuss how to access the treatment modules on the website and how to log in for subsequent video call sessions. The coach will email these participants a 'start-up' pack describing how to access the online materials.

Ten subsequent sessions (table 2) consisting of self-guided didactic content regarding problem-solving skills, video clips modelling these skills and exercises to practise the skills, will then be completed every 7 to 10 days

| Table 2 | TOPS-UK sessions (complete 10 in total) |
|---|---|
| Core sessions (complete all five) | Getting started and staying positive |
| | Steps of problem-solving |
| | Getting organised |
| | Staying in control |
| | Taking care of you |
| Tailored sessions (choose four) | Dealing with fatigue |
| | Managing fear and worry |
| | Controlling your anger and improving communication |
| | Listening, talking and reading non-verbal cues |
| | Social behaviour and joining a group |
| | Working with the school |
| Core final session | Bringing it all together |

TOPS-UK, Teen Online Problem-Solving, British version.

by the family and the adolescent with pABI. The online modules should take 30 min to complete, with the video call sessions taking 60 min. The parent will work together with the adolescent to complete the sessions. When the family has reviewed the materials for each session, the TOPS-UK coach will conduct a video call with the adolescent and parent who agreed to participate in the study. During this session they will review the online materials and practise the problem-solving skills using a problem identified by the family/adolescent. They will then plan the next session and agree a suitable time for the video call. Ideally, video calls will be held weekly, but this period can be increased as agreed through discussions between the coach and the family.

For all families allocated to TAU + TOPS-UK, the TOPS-UK coach will record details in the study database relating to intervention compliance for example, number and date of Skype sessions completed, length of each session, progress/engagement of adolescent and parent.

Completion of all 10 modules is expected to take each family 16 weeks in total. If the programme has not been completed after 16 weeks the intervention will be discontinued at that point. Study follow-up will proceed as if the programme had been completed.

### Sample size calculation
The study will aim to screen 20 potential participants at each site (n=100 in total), and aim to recruit a sample size of 25 participants from five sites. This should provide sufficient data to assess the feasibility and acceptability of the study. Although the sample size for the full RCT will be estimated based on the minimum clinically important difference for the BRIEF score[35] (5-points), the SD for this patient population is currently unknown. A sample size of 25 is considered a realistic target and would be sufficient to address the feasibility aims (ie, acceptability of questionnaires etc), as well as aiming to provide at least 12 participants reporting quantitative baseline data to calculate the required SD.[36]

### Data management
The CTU data management team is responsible for data management. Each participant will be allocated a unique trial number on consenting to participate and will be identified in all study-related documentation by the trial number and initials. A record of names and addresses linked to participants' trial numbers will be maintained by the research nurses at each site for administrative purposes and stored securely. This is an online study, with no requirement for data entry at CTU. Functions within the study website will remind participants to complete online measures, and flag up missing fields. However, participants will be able to progress through a questionnaire leaving data fields unfilled. The SQL Server database will be designed and maintained by the CTU data programming teams. Access by researchers will be password protected. In order to avoid problems with mislaid usernames and passwords, participants will access the web pages through links emailed to them by CTU. Once a web page has been completed by the participant it will be locked to prevent further data entry.

### Confidentiality
All data will be collected and managed in accordance with the Data Protection Act 1998. Each participant will be allocated a unique study number and will be identified in all study-related documentation by their study number and initials. All data will be entered on a password-protected SQL Server database and encrypted using a stored procedure. After all data cleaning has been performed and the database locked, anonymised data will be exported to the trial statistician.

### Data analysis plan
All randomised participants will be included in the analyses according to their randomised allocation, irrespective of adherence to treatment in the TOPS-UK arm or receipt of treatment in the TAU arm. No imputation of missing baseline or follow-up data will be performed. The study is not sufficiently powered to detect a significant treatment effect with regard to clinical or cost-effectiveness and thus a formal comparison will not be undertaken. The reported analyses will therefore be restricted to descriptive statistics on the outcome measures with appropriate point estimates (mean, median, percentage, etc), SD and 95% CIs for between group differences. For the questionnaire outcomes, approaches to missing individual items will be in accordance with the guidelines for missing item procedures for each questionnaire. Where no guidelines for individual missing items are available, the mean of the completed items will be used to replace missing items if 10% or fewer are missing. Statistical analyses will be

performed following final data cleaning and locking of the dataset. No interim analyses are planned. All analyses will be performed using Stata V.14 and will be performed by a statistician using a data set with treatment allocation masked.

## Assessment of trial feasibility

The primary feasibility outcomes of this study include: (i) number of participants at each stage of the study, at each site, with stages including: identified as potentially eligible, approached, consented to study, completed screening and baseline, randomised (with at least five participants being recruited at each site), completed treatment and completed outcome assessments; (ii) any difficulties experienced at sites that may affect their ability to recruit in future will be identified and reviewed; (iii) evaluation of appropriateness of the trial and economic evaluation methods and procedures; (iv) assessment of participants' willingness to be randomised to treatment allocation; (v) adherence to treatment assessed as number of sessions completed, a session will be recorded as complete if the participant has been through every page, reached the end and completed the coach web-linked session related to that material (with a target of participants completing at least five sessions) and (vi) attrition (with a target of at least 80% of participants completing follow-up assessments).

We will report the proportion of screened families who are found to be eligible, the proportion of eligible families who are recruited and randomised to their allocation and the proportion of randomised participants who provide outcome data at follow-up, with 95% CIs. Our sample size of 25 participants will allow us to estimate loss to follow-up (anticipated to be 20%) with a 95% CI of +/-13 percentage points. The further outcome is evaluation of the SD of the BRIEF-2 (parent) score for this patient population. This is the proposed primary outcome for the main trial and required for calculation of the sample size.[36]

## Measurement of outcomes

This feasibility study aims to evaluate all aspects of the proposed fully powered RCT and cost-effectiveness analysis including recruitment and retention numbers, and completion of outcome measures proposed for the main trial. Outcomes will be assessed by questionnaires and interviews. Table 3 provides a summary of outcome measures for this feasibility trial.

| Table 3 Summary of outcome measures | | | |
|---|---|---|---|
| **Outcome group** | **Outcome measure** | **Objective** | **Evaluation time point(s)** |
| Primary outcome (main trial) | Parent BRIEF-2 | 6 | Baseline, 17 weeks post-randomisation |
| Secondary outcomes (main trial) | Adolescent: RCADS, SDQ, CBQ, BRIEF-2, CHU-9D Adult: RCADS, SDQ, EQ-5D-5L*, PHQ-9*, GAD-7*, CBQ, CSRI | 1,10,11 | Baseline, 17 weeks post-randomisation |
| Measures of adherence (intervention group) | Participation in weekly Skype sessions. Record of login to intervention website (frequency, duration, progression) | 3,11 | Data recorded by coach. Data captured automatically by database throughout 16 week intervention period. |
| Intervention feasibility and acceptability (intervention group) | Qualitative interviews with families Intervention acceptability questionnaires (adolescent and parent) Qualitative interview with coach | 1–4, 6, 9 | End of trial 17 weeks post-randomisation End of trial |
| Experience of TAU (TAU arm) | Qualitative interviews with families | 15 | End of trial |
| Study acceptability (both groups) | Study participation questionnaires (adolescent and parent) Qualitative interviews (adolescent and parent) | 5 | End of trial |

*Measures relating to parental outcome.
BRIEF-2, Behaviour Rating Inventory of Executive Function, Second Edition; CBQ, Conflict Behaviour Questionnaire; CHU-9D, Child Health Utility 9D; CSRI, Client Service Receipt Inventory; EQ-5D-5L*, EuroQol 5 Dimensions 5 Levels Questionnaire; GAD-7*, Generalised Anxiety Disorder 7-item; PHQ-9*, Patient Health Questionnaire 9; RCADS, Revised Child Anxiety and Depression Scale; SDQ, Strengths and Difficulties Questionnaire; TAU, treatment as usual.

**Table 4** Trial schedule

| Study procedure | Screening | Baseline | 16week intervention period | Follow-up 17 weeks |
|---|---|---|---|---|
| Consent/assent (adolescent+/-parents) | X | | | |
| Consent (parents) | X | | | |
| Characteristics of ABI (CRF) | X | | | |
| Current medication (adolescent) | | X | | X |
| BRIEF parent-rating | X | | | X |
| Parents | | | | |
| Demographics, past medical history | | X | | |
| SDQ parent-rated | | X | | X |
| PHQ-9 | | X | | X |
| GAD-7 | | X | | X |
| CBQ | | X | | X |
| EQ-5D-5L | | X | | X |
| CSRI | | X | | X |
| RCADS parent version | | X | | X |
| Adolescents | | | | |
| Consent/assent (participants+/-parents) | X | | | |
| BRIEF-2 | | X | | X |
| RCADS | | X | | X |
| SDQ | | X | | X |
| CHU-9D | | X | | X |
| CBQ | | X | | X |
| Both parent and adolescent | | | | |
| Treatment adherence | | | | X |
| Treatment acceptability rating | | | | X |
| Study participation feedback | | | | X |
| Qualitative telephone interviews (all) | | | | X |

ABI, acquired brain injury; CBQ, Conflict Behaviour Questionnaire; CHU-9D, Child Health Utility 9D; CRF, case report form; CSRI, Client Service Receipt Inventory; BRIEF-2, Behaviour Rating Inventory of Executive Function, Second Edition; EQ-5D-5L, EuroQol 5 Dimensions 5 Levels Questionnaire; GAD7, Generalised Anxiety Disorder 7-item; PHQ-9, Patient Health Questionnaire 9; RCADS, Revised Child Anxiety and Depression Scale; SDQ, Strengths and Difficulties Questionnaire.

## Outcome measures

These are detailed in table 4. The proposed primary outcome for the main trial is the BRIEF-2,[34] a parent and self-report measure of everyday executive function skills. Secondary outcomes will include parent reports of their child's health and behaviour, using the Revised Child Anxiety and Depression Scale[37] (RCADS) and Strengths and Difficulties Questionnaire[38] (SDQ) to evaluate the adolescents' social, emotional and behavioural functioning.

A mixed methods approach (questionnaires and interviews) will be used to address outcomes relating to adolescent and parent acceptability of the intervention and outcome measures, coach's experience of supporting intervention delivery, parents' and adolescents' experiences of study participation, assessment of participants'

willingness to be randomised to treatment and complete outcome measures, evaluation of appropriateness of the trial and economic evaluation methods and procedures and adolescent and parent experiences of TAU (TAU arm).

Parental reports of their own and family quality of life, health, emotional functioning and family interactions will be evaluated using the EuroQol 5 Dimensions 5 Levels questionnaire[39] (EQ5D-5L), the Patient Health Questionnaire[40] (PHQ-9), the Generalised Anxiety Disorder[41] (GAD-7) questionnaire and the Conflict Behaviour Questionnaire[42] (CBQ).

Use of health and social care resources will be assessed through parental report on the Modified Client Service Receipt Inventory[43] (CSRI). Adolescents involved in the study will also complete questionnaires relating to their executive function skills (BRIEF-2[34] adolescent version), quality of life (Child health-related quality of life[44]; CHU-9D), emotional functioning (RCADS),[37] social, emotional and behaviour functioning (SDQ)[38] and their family interactions (CBQ).[42]

The EQ-5D-5L and CHU-9D also provide data on quality-adjusted life-years (QALYs) which can be used in cost-effectiveness analyses.

## Follow-up questionnaires

Adolescents and their parents in both treatment arms will be sent an email asking them to complete their online follow-up outcome measures at 17 weeks post-randomisation. In the event of non-completion, at least two reminders will be sent by email, followed by a reminder telephone call from the site research team if required. As with the baseline measures, participants will be given the option (at consent/assent) to complete the online follow-up measures with telephone support from a member of the research team. If participants do not complete the follow-up measures within 4 weeks they will receive a 'last chance' email with a reminder of the study link. A £15 gift voucher will be sent to each family following completion of the follow-up questionnaires.

It is acknowledged that a definitive trial would require longer follow-up than this feasibility study will provide, but the feasibility trial will inform the acceptability of outcome measures and the experience of participating in the trial. The power calculation for a definitive trial with longer follow-up will account for a higher level of attrition than this feasibility trial.

## QUALITATIVE INTERVIEWS

A single qualitative researcher will conduct semi-structured qualitative process evaluation interviews by telephone or video call with all participants (including those who have not completed follow-up questionnaires but who have not formally withdrawn from the study), research nurses and the study coach. PIS relating to the qualitative interviews will be emailed in advance and consent will be sought at the beginning of the interview by the qualitative

researcher. Adolescent and parent participants will be offered the option of being interviewed together or separately. Interviews will be conducted with the help of an agreed topic guide (see online supplementary appendix) and are expected to last between 20 to 60 min. Interviews will be audio-recorded and transcribed verbatim. The transcribed interview data will be fully anonymised and any demographic data about participants will be stored separately. Qualitative interview data will be managed using a computer software package such as Nvivo 11 and thematically analysed.[45] The analysis and results will be checked/validated by a second qualitative researcher. The following interviews will be conducted:

1. **All participants** (both groups). These interviews will allow participants to describe their whole experience of participating in the study including their willingness to be randomised and their experiences and acceptability of outcome measures, what worked well and what less well. This will cover aspects such as clarity of PIS, acceptability and ease of completing the online consent process, ease of completing online questionnaires and the parental support required the adolescent to participate and remain engaged. The interviews will explore adolescent and parent experiences of TAU or TAU + TOPS-UK, including parent's perceptions of the impact of TAU/TAU + TOPS-UK on their ability to support the adolescent, adolescent's and parent's perceptions of the impact of TAU/TAU + TOPS-UK on adolescent's ability to stay positive, solve problems, be organised, control their emotions and look after themselves (self-care), and participants' views on potential improvements to TAU/TAU + TOPS-UK. In addition, participants in the TAU + TOPS-UK will be asked about their experience of working with the coach via video calls. This information will be helpful when planning the full RCT and implementation studies. After completion of the interview, each family will be sent a £15 gift voucher as a token of appreciation.

2. **Interview with TOPS-UK coach**. A single interview will explore the coach's experiences of supporting the TOPS-UK intervention. The interview will be held by telephone, or face-to-face, once every participant allocated to the intervention arm has completed the programme.

3. **Interviews with research nurses.** All research nurses involved with the study at the five participating sites will be invited to participate in a single interview to explore their experiences of supporting families through the consent process and any support required to complete baseline or follow-up measures. Feedback on methods of contact with families, number of contact attempts made, how much support was required from adolescents and/or parents, barriers to recruitment and suggestions for future studies will also be sought.

Participants who withdraw from the study, participants who register their interest but do not complete the study consent forms and participants who discontinue the intervention and who do not complete the follow-up

questionnaires (ie, non-adherent but not explicitly withdrawn), will be invited to complete an anonymous online feedback survey. The survey will invite participants to comment on any aspects of the study that they found difficult, any aspects of the study that they liked and suggestions on how the study can be improved.

## ECONOMIC EVALUATION

This feasibility study will be used to develop a framework for a subsequent, policy-relevant, cost-effectiveness analysis to be undertaken alongside a future RCT. Economic evaluation methods will be developed and assessed regarding the collection of resource use, cost and outcome data. Data on resource use associated with the set-up and delivery of the TOPS treatment will be collected at the participant and coach level for example, coach contact and non-contact time per participant, equipment and consumable costs, training and supervision requirements for the TOPS-UK coach. Data regarding service use will be collected from all participants' parents using a version of the CSRI specifically modified for this population. This includes use of mental health, community rehabilitation and neuropsychology and educational psychology services and has been previously trailed in this population (Wilson *et al*, in preparation). In addition to informing the development of the economic evaluation methods, this will provide a profile of the resources and services that constitute TAU. The study will also consider the most appropriate manner in which to capture data on health-related quality of life for use in the estimation of the cost per QALY of the TOPS-UK treatment. The feasibility of participants self-completing the CHU-9D and parents completing the EQ-5D-5L, will be assessed.

In addition to this primary economic outcome, the appropriateness of considering the cost-per unit change on other relevant indicators of health status collected in the baseline and follow-up phases will be explored.

## SAFETY REPORTING

The risks associated with participating in this study are considered minimal. There is a slight chance for those in the intervention group that raising awareness of injury-related cognitive or behavioural problems through communication and problem-solving might increase family burden and possibly contribute to conflict between family members. However, the purpose of the intervention is ultimately to equip families with skills to handle these difficulties by learning how to change the way they solve problems and talk with one another, and the TOPS-UK coach supporting the online intervention will be trained to handle any emerging problems. Should any issues arise, the coach will have access to a qualified clinical psychologist to provide further support and advice. There is therefore no requirement to report non-serious adverse events in this study, although serious adverse events (SAEs) will be monitored.

SAEs may be reported by clinicians or researchers at site, the TOPS-UK coach, by participants themselves or by any other informant. Adverse event data will be monitored by the Trial Steering Committee (TSC) to ensure safety. The TSC includes an independent statistician, psychologist, paediatrician and patient and public representative and will meet on approximately four occasions. All suspected SAEs will be reported within 24 hours of discovery to the CTU who will notify the Chief Investigator. All SAEs will be followed up until resolution.

## PARTICIPANT AND PUBLIC INVOLVEMENT

A PPI group including patients and the CI was convened at an early stage of grant development and trial design and continued to meet during study set-up. The group gave feedback on the patient-facing materials including age-appropriate PIS, informed consent forms and qualitative interview topic guide content. A lay-representative who is Director of Services and Innovation at the national Child Brain Injury Trust (CBIT) is a co-applicant to the grant, and a member of both the Trial Management Group (TMG) and TSC. During the recruitment period, this lay-representative has been liaising with local CBIT representatives and PICs to identify potentially eligible families. A study twitter account will post trial updates to promote public engagement with the study. The twitter site currently follows and is followed by national brain injury charities disseminating trial updates to a wide audience.

Another PPI representative (parent of a child with pABI) is a member of the TMG, regularly attending meetings and freely contributing to the discussions. Both members provide advice and suggest solutions to problems encountered during the trial, with particular expertise in barriers to recruitment and communication issues within families with a teenager with pABI. This expertise has been cascaded to site staff through a research nurse forum.

The contribution of PPI members within the TMG and TSC will be valuable during analysis, interpretation and dissemination of the study results. Specifically, PPI members will be asked to contribute to data interpretation, assess whether the feasibility study has met its objectives and support different approaches to dissemination of the study. They will also be asked to comment on potential changes for the main trial. The Director of Services and Innovation at CBIT will also support wider PPI in the development of the main trial.

## STUDY MANAGEMENT AND OVERSIGHT

The study sponsor organisation is the Royal Devon and Exeter NHS Foundation Trust, Barrack Road, Exeter EX2 5DW. Day-to-day trial management is administered through the UKCRC-registered Peninsula Clinical Trials Unit at Plymouth University. A Trial Management Group including the Chief Investigator, CTU trial managers,

trial statistician and other personnel relevant to the study (eg, clinicians, CTU data manager, patient and Sponsor representatives) will meet regularly (usually monthly) throughout the duration of the trial to oversee practical management of the trial.

A TSC, chaired by an independent member, will oversee the conduct and safety of the trial, ensuring that milestones are achieved and general scientific probity is maintained. A Data Monitoring Committee was not required for this feasibility study.

### Ethics and dissemination

The study will be undertaken at acute NHS Trusts, subject to appropriate Research Ethics Committee (REC) and Health Research Authority approvals. The trial will be conducted in accordance with the protocol, the principles of the Declaration of Helsinki and International Conference on Harmonisation (ICH) Good Clinical Practice (GCP). Any amendments of the protocol will be submitted to the REC for approval. On request the Chief/Principal Investigator should make available relevant trial-related documents for monitoring and audit by the Sponsor or the relevant Research Ethics Committee.

The study team will prepare a plain English summary of the study results which will be sent to the study participants as soon as possible after the end of the study. The final results of the study will be disseminated via presentations at appropriate scientific meetings and conferences and publication in appropriate peer-reviewed journals. The data from this study will be used to inform the design and accompanying grant application for a fully powered RCT should the study be considered feasible.

## DISCUSSION

In our opinion, TOPS-UK has genuine potential to have a profound and positive effect on the lives of thousands of adolescents with brain injury. The intervention itself already exists and has demonstrated efficacy in the USA, including gains in executive function (planning, problem-solving), reductions in behaviour and mood difficulties and reductions in family burden and stress. TOPS-UK urgently requires testing in a rigorous multicentre randomised trial in the UK, allowing a definitive test of its clinical and cost-effectiveness.

The importance of the current study is to determine if a large RCT is feasible (eg, recruitment, access to resource use data, appropriate outcome measures etc). In addition, although the study is not powered to detect effectiveness, individual patients participating in the study might gain direct benefit from the treatment (TAU + TOPS) in terms of increased executive function skills, reduced behaviour and mood difficulties, and improved quality of life. The feasibility study might also have some implications on clinical and public health practice by raising awareness of pABI, the need for effective treatment, and the variability of TAU (via the dissemination workshop and reports), and improve the measurements of quality of life and

health status in adolescents with pABI (via conferences). If the current study demonstrates feasibility, outcomes will be used to inform the development of a fully powered phase III RCT to examine effectiveness and cost-effectiveness of TOPS-UK.

If the phase III RCT is successful, then the clinical and public health practice developments are potentially far reaching. For example, in addition to the direct benefits to the adolescent and his or her family, TOPS-UK might also prevent considerable NHS costs in the future by reducing healthcare service use. Furthermore, given the increased risks of criminal behaviour and poor educational and vocational outcomes following pABI, often associated with poor executive functioning, TOPS-UK might have wider social benefits for these individuals, and society as a whole. In terms of potential impact on local policy-making and improvement in service delivery, TOPS-UK is potentially very cheap to deliver because it is web-based and involves regular but brief professional support that is delivered remotely (via video calls). Furthermore, given its web-based delivery that can be accessed at a time that is convenient for the adolescent and their family, TOPS-UK is less disruptive than clinic appointments, which are often delivered during school-hours. This is an important consideration for an individual with pABI because, typically, the adolescents have already missed a significant amount of school and struggle academically. By enabling free use of the materials to the NHS, clinical services might routinely provide TOPS-UK to adolescents with pABI and their families, meeting the needs of multiple families simultaneously. TOPS-UK might also provide a first-line of treatment for families experiencing distress and, therefore form part of a 'stepped-care' model of service delivery such that, families who continue to experience significant difficulties, or who are at high risk, can be referred to specialist services in a timely manner. The ultimate aim of the research programme is to provide an evidence-base for effective neuropsychological interventions to be recommended by the Department of Health guidelines (eg, NICE) for survivors of pABI.

**Author affiliations**

[1]University of Exeter, Exeter, UK
[2]Cincinnati Children's Hospital Medical Center, Cincinnati, UK
[3]Peninsula Clinical Trials Unit, Plymouth University, Plymouth, UK
[4]Peninsula Clinical Trials Unit, University of Plymouth, Plymouth, Devon, UK
[5]University of Exeter Medical School, Institute of Health Research, Exeter, UK
[6]Patient Representative, Exeter, UK
[7]Psychology, University of Exeter, Exeter, UK

**Acknowledgements** The authors are grateful for the support of the study sponsor (Royal Devon and Exeter NHS Foundation Trust) and the South West Peninsula NIHR Clinical Research Network. We acknowledge the contribution made by Dr Richard Tomlinson, Consultant Paediatrician, Royal Devon and Exeter NHS Foundation Trust who was a grant co-applicant and for his support during study set-up. We are also indebted to Theresa Pass, former Director of the Child Brain Injury Trust who was a grant co-applicant, and to Louise Wilkinson and members of the Child Brain Injury Trust for their continued support and advice during recruitment and study delivery.

**Contributors** All authors except AJ and SH were co-applicants on the NIHR RfPB grant application and as such were involved in the design of this feasibility study. All authors contributed to successive drafts of this paper. JL was the CI for

the first year of the study and led on developing the intervention, finalising the protocol and drafting this manuscript. SW designed the study intervention and contributed to study design. JV contributed to the study design and oversaw trial management. AJ is the trial manager, responsible for the day-to-day running of the trial and contributed to drafts of this manuscript. FW is the trial statistician and provided expertise in the overall design of the trial. AH was responsible for the design and analysis of the economic evaluation component. JS was responsible for the design and analysis of the qualitative component. TF contributed to study design and provided advice and guidance on study delivery. SH contributed to trial management from a PPI perspective. AA was the lead grant applicant and took over as CI from JL for the second year of the study, providing clinical expertise and drafting and revising this manuscript.

**Funding** This paper presents independent research funded by the National Institute for Health Research (NIHR) under its Research for Patient Benefit (RfPB) Programme (Grant Reference Number PB-PG-0614-34081). The views expressed are those of the author(s) and not necessarily those of the NHS, the NIHR or the Department of Health and Social Care. The funder has no role in study design or the collection, management, analysis or interpretation of data.

**Competing interests** None declared.

**Ethics approval** This study has been granted ethical approval by the South West-Exeter Research Ethics Committee (ref 17/SW/0083).

**Provenance and peer review** Not commissioned; externally peer reviewed.

**Data sharing statement** After the end of the study, information collected may be made available as an anonymised participant level data set to other researchers under an appropriate data sharing agreement.

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
