## [Reviewer comments · BMJ Open]

ARTICLE DETAILS

TITLE (PROVISIONAL)	The clinical and cost effectiveness of Teen Online Problem-Solving for adolescents who have survived an acquired brain injury in the UK: Protocol for a randomised, controlled feasibility study (TOPS-UK)
AUTHORS	Limond, Jenny; Wade, Shari L.; Vickery, Patricia; Jeffery, A; Warren, Fiona; Hawton, Annie; Smithson, Janet; Ford, Tamsin; Haworth, Sarah; Adlam, Anna-Lynne

VERSION 1 - REVIEW

REVIEWER	Mark Holloway Head First UK
REVIEW RETURNED	28-Jan-2019

GENERAL COMMENTS	The investigation of paediatric ABI, the impact this has over time, how executive skills are affected and the functional impact this has, leading to less favourable life outcomes, is very important. I am pleased that a team wish to examine this. I understand that this is a feasibility study to look at a larger RCT and whether this is possible and this programme may well address this question however there are other issues at play. You may consider that some of what I raise here has little relevance for the research suggested, but I think that it has some perhaps. 1. TAU, usually, means no service whatsoever (or a very poor one, not a well funded interdisciplinary, neuropsychology led community rehabilitation package augmented by adequate rehabilitation assistants, managed by a brain injury case manager, working across settings and time, involving family, friends and school), hence what may be being measured is the receipt of any service, not the efficacy of the TOPS scheme which seems to be accepted as effective without question.2. The follow up period for effectiveness of rehabilitation/support for executive impairment issues (functional, what happens in the real world) should be years not one week post intervention. That cannot be measured here by what is planned (as noted, I know this is about feasibility for a larger study, not effectiveness of intervention to create genuine self-initiated functional changes over years)3. Bearing in mind potential drop out, very significant and
--

unknowable heterogeneity of participant (pre and post injury), the inclusion of non-traumatic injury (potential for very focal injury to be being measured against very diffuse axonal injury, "ill" children to be measured against "injured" children), the fact that some participants will be experiencing natural recovery, the fact that executive impairment takes time to develop post injury (or more accurately time to manifest), that children are at complicated stages of development in terms of their executive skills pre-injury, that parents approach parenting differently and so what skills they expect their offspring to have (what responsibilities they anticipate they will shoulder) are very different, therefore, n=50 looks a little light on the ground. I think that this is a limitation.

4. The use of different points of contact may mean that treatment adherence/effectiveness is a function of the person administering the treatment.

5. Parents are learning about executive impairment at the time this takes place. Participants may be relatively recently injured, parents are learning what invisible changes have taken place to their offspring during the trial. Therefore, rather than supporting online problem solving (functionally) the programme may in fact serve to support the identification, naming and explaining of post-injury sequelae, which, in itself, gives parents and young people the information that they sorely lack and which research identifies as being a significant issue. Again, this is not TOPS being rated, it is any service, any information, any way of structuring knowledge and understanding against the alternative which is nothing.

6. The supporting literature does not have a lot about how decision making is supported, what are problem solving strategies and, sorry to go on about it again (!) whether these strategies are applied, successfully, in the moment they are required not in the abstract. This is relevant, there are claims within the research about potential reductions in health and social care costs. To be sure of this one would need to be measuring whether concrete, real world examples of problem solving had been improved and that this led to changed outcomes in terms of functioning. I think that SARIASLAN, A., SHARP, D. J., D'ONOFRIO, B., LARSSON, H. & FAZEL, S. 2016. Long-Term Outcomes Associated with Traumatic Brain Injury in Childhood and Adolescence: A Nationwide Swedish Cohort Study of a Wide Range of Medical and Social Outcomes. PLoS Medicine, 13. is helpful as is the work of Knox and Douglas. I think that the team could augment their literature (and beef up their argument) by inclusion of research that recognises just how information is a key need following ABI and one which is rarely met. I think that the literature could be expanded to include more on the parental experience of childhood ABI and noting the tension in rehabilitation approaches behind more mainstream cognitive rehabilitation and neurofunctional approaches which recognise the impact insight etc has on the ability to self initiate compensatory strategies. Kreutzer et al have written extensively about their manualised approach to working with relationships post ABI, I think that the TOPS system has things in common (albeit Kreutzer's interventions are face to face). Bombardier has written about the use of telephone interventions for depression post ABI, that may be useful?

Overall, I think that this is an ambitious project with a lot of measures. I am very pleased to see the qualitative component, the acceptability of the intervention and the experience of receiving it will

	likely throw up some interesting observations and themes. It is very useful that this has been included and I hope that the guide for the qualitative interviewers allows for a broad and fare-ranging discussion. I can foresee a separate and useful paper on the themes generated by these conversations. Being the parent of a child with a brain injury is an astonishingly taxing and difficult experience, the more those voices are heard the better. I will be interested to see where this heads. Thanks for letting me review this, apologies if some of my comments are less than relevant for the task in hand, hopefully not just my hobby-horse but something of use to consider re the effectiveness over time, functional changes etc.
--	--

REVIEWER	Fergus Gracey University of East Anglia, UK I am an honorary consultant neuropsychologist at one of the recruitment sites for this trial, although have not been involved at all in any study-related tasks.
REVIEW RETURNED	22-Feb-2019

GENERAL COMMENTS	The authors present the protocol of a study testing the feasibility of an RCT of an established online intervention for adolescents with an acquired brain injury (TOPS). The intervention has been developed and evaluated in the USA and the current protocol aims to establish feasibility of a trial of an adapted, UK version (TOPS-UK). The planned study will extend from traumatic brain injury (as evaluated in the US) to include a wider range of acquired injury. The background correctly delineates the need for evidence based interventions for supporting children with an acquired brain injury and presents a rationale for this type of intervention (addressing higher level executive skills) which are especially relevant at this developmental stage, and a known area of need for adolescents who sustained a brain injury. The protocol is mostly very well described and in keeping with the SPIRIT guidelines. There is a small number of issues where further clarification or detail may be of help, as follows:  1. The The study objectives, as set out in Table 1, are broadly appropriate for a feasibility RCT. However, I think some of these could be articulated in a more specific way e.g. how is clinician ability to identify participants going to be measured / reported, and how does this differ from item 3? Item 3 - is recruitment rate important to establish? what numbers might be required in a given time period? I wonder if objectives could be rephrased as questions (e.g. Do the trial and economic evaluation methods and procedures yield the information required? or for Item 12- how many complete and analysable datasets are yielded, and are there any items or measures that show high / unacceptable levels of missing data 2. Related to this there is a lack of specific criteria against which to establish feasibility. The authors should be clear in stating criteria for feasibility where these are known, or if not known be clear in stating their intention to gather information that will inform the parameters of the full trial. 3. It would be helpful to know if the authors intend to establish what
---

	constitutes TAU for each participant, or establish a means of classifying interventions for a future trial (given wide variability in the type and availability of services for children with BI)? 4. 5 UK sites are mentioned. A little more detail regarding the nature of these sites would be helpful. In addition, are these sites the ones that will recruit to a full trial? Were they selected in order to be representative of the range of types of service which might be included in a future trial? 5. It is noted that sites that fail to meet a recruitment target might not be included in a future trial. Given likely challenges with future recruitment this might not be the most helpful approach to take. Will the process measures allow for identification of barriers to recruitment and ways of improving recruitment in the future full trial, rather than simply disqualifying potential recruitment sites? In addition, no criteria is provided for 'poor recruitment'. 6. page 5 - how will the PI establish presence of EF difficulties? 7. page 5 - ICH needs writing out in full 8. top of page 6, Good Clinical Practice training might need an explanation for non-UK readers. 9. page 10, assessment of trial feasibility - as noted previously there is a lack of specific criteria against which feasibility might be assessed e.g. how will 'poor recruitment' be defined? Will measures of uncertainty of estimates of feasibility objectives (such as recruitment rate) be calculated i.e. 95% CI's? The authors state the sample size will be sufficient to establish feasibility - however it is not possible to establish if this is the case from the information provided, for example what sample size is required to estimate recruitment rate at an acceptable level of uncertainty? 10. Table 3 - adult-completed secondary outcomes - can those that are child proxy measures be separately labelled to those which are measures of parent outcome? 11. page 14 - the line 'The contribution of PPI members ...' - this is important, but seems somewhat vague, especially after the preceding paragraph setting out specific PPI contributions. Can the authors add specific plans regarding this valued PPI input in the study going forward? 12. There is a lack of follow up time point - this is a significant weakness given the need for a follow up timepoint in a full trial, and also because longer term outcomes are especially important given the presence of brain injury within the developmental context. Some recognition of protocol limitations would be helpful in the discussion. 13. Online appendix - there is a bullet point without accompanying information
--	--

VERSION 1 – AUTHOR RESPONSE

Reviewer: 1

Reviewer Name: Mark Holloway

Institution and Country: Head First UK

Please state any competing interests or state 'None declared': None declared

Please leave your comments for the authors below

The investigation of paediatric ABI, the impact this has over time, how executive skills are affected and the functional impact this has, leading to less favourable life outcomes, is very important. I am pleased that a team wish to examine this.

I understand that this is a feasibility study to look at a larger RCT and whether this is possible and this programme may well address this question however there are other issues at play. You may consider that some of what I raise here has little relevance for the research suggested, but I think that it has some perhaps.

1. TAU, usually, means no service whatsoever (or a very poor one, not a well funded interdisciplinary, neuropsychology led community rehabilitation package augmented by adequate rehabilitation assistants, managed by a brain injury case manager, working across settings and time, involving family, friends and school), hence what may be being measured is the receipt of any service, not the efficacy of the TOPS scheme which seems to be accepted as effective without question.

We agree with the reviewer that the provision of TAU may vary widely in terms of the nature of services provided and quality of services, both across the trial sites and across the UK in general. A potential definitive trial would be pragmatic in nature, and based on effectiveness rather than efficacy of the intervention. Indeed, there is indirect evidence of the efficacy of the intervention (relative to an internet resource comparison arm) from the TOPS trials conducted in the USA (e.g., Wade et al., 2010; 2012). The aim of the definitive trial, therefore, is to compare TOPS-UK with the routinely available care that participants would otherwise receive, rather than comparing versus a standardised comparator group (as this would not reflect the true situation with regard to treatment for this patient group). For a definitive trial we would seek to describe in detail the TAU as received by participants in both arms (as TOPS-UK participants would also receive TAU as provided in their location). A definitive trial would also seek to investigate any differential treatment effect across sites (which could be due to differential effectiveness of TAU, among other explanations), although such a trial would not be powered to detect such a differential treatment effect.

We have added the following text to page 5 of the manuscript: "These sites were selected to be representative of the range of types of services which might be included in a future trial, with TAU ranging from no provision through to long-term out-patient multi-disciplinary teams."

2. The follow up period for effectiveness of rehabilitation/support for executive impairment issues (functional, what happens in the real world) should be years not one week post intervention. That cannot be measured here by what is planned (as noted, I know this is about feasibility for a larger study, not effectiveness of intervention to create genuine self-initiated functional changes over years)

We agree with the reviewer that a definitive trial would require longer follow-up than this feasibility study will provide. However, we feel that the follow-up period, although shorter than required for a definitive trial, will allow us to receive valuable feedback from participants regarding the acceptability

of outcome measures and their experiences of participating in the trial. We acknowledge that retention achieved after a short follow-up period may not be achieved in the longer term; in view of this we would ensure that the power calculation for a definitive trial, with longer follow-up, accounted for attrition at a higher level than will be observed in this feasibility trial.

The following text has been included on page 14 of the manuscript: "It is acknowledged that a definitive trial would require longer follow-up than this feasibility study will provide, but the feasibility trial will inform the acceptability of outcome measures and the experience of participating in the trial. The power calculation for a definitive trial with longer follow-up will account for a higher level of attrition than this feasibility trial."

We have also included the following study limitation on page 2 of the revised manuscript: "This feasibility study was conducted over a short time period thus limiting the opportunity for a long-term follow-up. The definitive trial would include a follow-up period of at least 6 months."

3. Bearing in mind potential drop out, very significant and unknowable heterogeneity of participant (pre and post injury), the inclusion of non-traumatic injury (potential for very focal injury to be being measured against very diffuse axonal injury, "ill" children to be measured against "injured" children), the fact that some participants will be experiencing natural recovery, the fact that executive impairment takes time to develop post injury (or more accurately time to manifest), that children are at complicated stages of development in terms of their executive skills pre-injury, that parents approach parenting differently and so what skills they expect their offspring to have (what responsibilities they anticipate they will shoulder) are very different, therefore, n=50 looks a little light on the ground. I think that this is a limitation.

Thank you for this comment. The original sample in the study was planned to be 50, however, for pragmatic reasons, following advice from the Trial Steering Committee and the Trial Management Group, this was reduced to 25 participants. The revised protocol with the reduced sample size was submitted for ethical review on 6 May 2018 and received a favourable opinion on 12 June 2018. The manuscript submitted to BMJ Open was based on the amended study protocol and, therefore, should have stated the sample size as 25 participants. We have amended the revised manuscript to indicate the planned sample size of 25 participants and we apologise for not correcting this error prior to the original submission of the manuscript.

For the purpose of this feasibility trial, we are not aiming to have the power to detect a between group difference in any of the clinical outcome measures. We, therefore, consider that a sample size of 25 participants is sufficient to achieve the feasibility aims of the study (e.g., Julious, 2005), as set out in the Introduction. In particular, we wish to assess loss to follow-up and estimate standard deviations to inform the sample size calculations for a definitive trial.

We acknowledge that there will be heterogeneity among adolescent participants and their parents, and that we cannot capture the full range of clinical and family situations in a feasibility study. The definitive trial will be sufficiently powered to detect between group differences on the clinical outcome

measures, and we can examine sub-group analyses (e.g., types of brain injury) and the effects of potential moderator variables (e.g., age at injury, time since injury, parenting) on outcomes in the definitive trial.

4. The use of different points of contact may mean that treatment adherence/effectiveness is a function of the person administering the treatment.

For the feasibility trial, only one trained TOPS-UK coach will deliver the intervention via video-conference facilities (e.g., Skype). This has been clarified on page 8 of the manuscript, with the following sentence included: "One coach will be trained to deliver the intervention to all of the participants in the intervention arm".

For the definitive trial, if multiple coaches are administering the intervention, then all coaches will be trained to the same standards, and treatment adherence procedures will be followed (e.g., working from a manual, clinical supervision with a TOPS-trained clinician, completion of treatment adherence measures).

5. Parents are learning about executive impairment at the time this takes place. Participants may be relatively recently injured, parents are learning what invisible changes have taken place to their offspring during the trial. Therefore, rather than supporting online problem solving (functionally) the programme may in fact serve to support the identification, naming and explaining of post-injury sequelae, which, in itself, gives parents and young people the information that they sorely lack and which research identifies as being a significant issue. Again, this is not TOPS being rated, it is any service, any information, any way of structuring knowledge and understanding against the alternative which is nothing.

We agree that for an efficacy trial it is important to control for the general benefits of an intervention. Indeed, our co-author and collaborator, Prof Shari Wade, and her team have demonstrated evidence of the efficacy of TOPS (albeit in the USA) using randomised controlled trials with participants randomised to TOPS or an internet resource comparison arm (e.g. psychoeducation without specific problem-solving strategies). We have cited some of these studies in the Introduction to the manuscript (e.g., Wade et al., 2010; 2012).

The purpose of the current study is to establish whether a definitive clinical and cost-effectiveness trial is feasible in UK. As mentioned in response to comment 1 above, the aim of the definitive trial, therefore, will be pragmatic in nature and will compare TOPS-UK with the routinely available care that participants would otherwise receive.

6. The supporting literature does not have a lot about how decision making is supported, what are problem solving strategies and, sorry to go on about it again (!) whether these strategies are applied, successfully, in the moment they are required not in the abstract. This is relevant, there are claims within the research about potential reductions in health and social care costs. To be sure of

this one would need to be measuring whether concrete, real world examples of problem solving had been improved and that this led to changed outcomes in terms of functioning. I think that SARIASLAN, A., SHARP, D. J., D'ONOFRIO, B., LARSSON, H. & FAZEL, S. 2016. Long-Term Outcomes Associated with Traumatic Brain Injury in Childhood and Adolescence: A Nationwide Swedish Cohort Study of a Wide Range of Medical and Social Outcomes. *PLoS Medicine*, 13. is helpful as is the work of Knox and Douglas. I think that the team could augment their literature (and beef up their argument) by inclusion of research that recognises just how information is a key need following ABI and one which is rarely met. I think that the literature could be expanded to include more on the parental experience of childhood ABI and noting the tension in rehabilitation approaches behind more mainstream cognitive rehabilitation and neurofunctional approaches which recognise the impact insight etc has on the ability to self initiate compensatory strategies. Kreutzer et al have written extensively about their manualised approach to working with relationships post ABI, I think that the TOPS system has things in common (albeit Kreutzer's interventions are face to face). Bombardier has written about the use of telephone interventions for depression post ABI, that may be useful?

Thank you for this comment. We have added the following additional information regarding the conceptual/theoretical basis for the TOPS intervention and the broader literature addressing problem-solving treatments for brain injury sequela to the Introduction (see page 3 and 4): “The development of online family problem-solving treatment for pediatric TBI was predicated on evidence of the reciprocity between child recovery and parental psychological well-being and informed by input from survivors of TBI and their families. Problem-solving therapy provides a systematic approach for coping more effectively with life’s challenges by creating a positive mind set and developing, implementing, and evaluating solutions to problems. Online family problem-solving treatment simplifies the ABCDEs of problem solving: defining the Aim, Brainstorming, Choosing the best strategy to implement, Doing it (creating a concrete, step by step plan), and Evaluating whether it worked. Given the focus on the family rather than the individual, TOPS also focuses on communication skills and collaborative problem solving. For adolescents with executive function and problem-solving deficits following TBI, the steps of problem solving also provide an executive function heuristic. Additionally, the program teaches stress management and metacognitive strategies such as stopping to think before acting and self-monitoring that can serve to promote self-regulation. Similar to the face-to-face, multifamily Brain Injury Family Intervention, online modules provide didactic information regarding the cognitive and behavioral consequences of TBI, training in problem-solving, communication, and self-awareness/regulation, and exercises to reinforce understanding. After completing the online modules on their own, families meet with a trained therapist to implement to the problem-solving process around an Aim (goal) that they have identified. Thus, the online problem-solving program was grounded in the scientific literature, tailored to address the unique neurobehavioral and family consequences of TBI, and refined through stakeholder input. Problem-solving therapy has also been trialed as a telephone-based intervention for adults with mild TBI and with families of adults with more significant brain injury, suggesting its potential utility across age groups and modes of delivery.”

As described, although TOPS provides some psychoeducation regarding brain injury, that is not its major thrust and, in fact, online information/psychoeducation regarding TBI has served as a control condition in the trials in the USA. For this reason, we have not added additional background regarding the need for information acutely following brain injury.

The reviewer also commented on the “need to be measuring whether concrete, real world examples of problem solving had been improved and that this led to changed outcomes in terms of functioning”.

As described in the Introduction (page 3 and 4), research to date has demonstrated improvements in executive function (as measured using the BRIEF, which assesses self and proxy reports of 'everyday' problem-solving and executive functioning), child behaviour, parental depression, and family-child conflict following TOPS compared to an internet resource comparison arm. To continue to capture improvements in problem-solving in the real-world, the BRIEF-2 is also the planned primary outcome for the clinical and cost effectiveness definitive trial.

7. Overall, I think that this is an ambitious project with a lot of measures. I am very pleased to see the qualitative component, the acceptability of the intervention and the experience of receiving it will likely throw up some interesting observations and themes. It is very useful that this has been included and I hope that the guide for the qualitative interviewers allows for a broad and fare-ranging discussion. I can foresee a separate and useful paper on the themes generated by these conversations. Being the parent of a child with a brain injury is an astonishingly taxing and difficult experience, the more those voices are heard the better.

Thank you. We agree with the reviewer that the qualitative component of the feasibility study will make an important contribution to the field and will provide an opportunity for multiple 'voices' to be represented.

I will be interested to see where this heads. Thanks for letting me review this, apologies if some of my comments are less than relevant for the task in hand, hopefully not just my hobby-horse but something of use to consider re the effectiveness over time, functional changes etc.

Reviewer: 2

Reviewer Name: Fergus Gracey

Institution and Country: University of East Anglia, UK

Please state any competing interests or state 'None declared': I am an honorary consultant neuropsychologist at one of the recruitment sites for this trial, although have not been involved at all in any study-related tasks.

Please leave your comments for the authors below.

The authors present the protocol of a study testing the feasibility of an RCT of an established online intervention for adolescents with an acquired brain injury (TOPS). The intervention has been developed and evaluated in the USA and the current protocol aims to establish feasibility of a trial of an adapted, UK version (TOPS-UK). The planned study will extend from traumatic brain injury (as evaluated in the US) to include a wider range of acquired injury. The background correctly delineates the need for evidence based interventions for supporting children with an acquired brain injury and presents a rationale for this type of intervention (addressing higher level executive skills) which are

especially relevant at this developmental stage, and a known area of need for adolescents who sustained a brain injury.

The protocol is mostly very well described and in keeping with the SPIRIT guidelines. There is a small number of issues where further clarification or detail may be of help, as follows:

1. The study objectives, as set out in Table 1, are broadly appropriate for a feasibility RCT. However, I think some of these could be articulated in a more specific way e.g. how is clinician ability to identify participants going to be measured / reported, and how does this differ from item 3? Item 3 - is recruitment rate important to establish? what numbers might be required in a given time period? I wonder if objectives could be rephrased as questions (e.g. Do the trial and economic evaluation methods and procedures yield the information required? or for Item 12- how many complete and analysable datasets are yielded, and are there any items or measures that show high / unacceptable levels of missing data

Thank you. Table 1 has been amended with the study objectives rephrased as questions and using more specific language/criteria where appropriate.

2. Related to this there is a lack of specific criteria against which to establish feasibility. The authors should be clear in stating criteria for feasibility where these are known, or if not known be clear in stating their intention to gather information that will inform the parameters of the full trial.

Thank you for this helpful comment. We have revised the phrasing in Table 1 to indicate the information that we will gather as part of the study objectives. We have also revised the section "Assessment of trial feasibility" on page 11 of the manuscript to specify the success criteria for the feasibility study. The following text has been revised: "The primary feasibility outcomes of this study include: (i) number of participants at each stage of the study, at each site, with stages including: identified as potentially eligible, approached, consented to study completed screening and baseline, randomised (with at least five participants being recruited at each site), completed treatment, and completed outcome assessments; (ii) any difficulties experienced at sites that may affect their ability to recruit in future will be identified and reviewed; (iii) evaluation of appropriateness of the trial and economic evaluation methods and procedures; (iv) assessment of participants' willingness to be randomised to treatment allocation; (v) adherence to treatment assessed as number of sessions completed, a session will be recorded as complete if the participant has been through every page, reached the end and completed the coach web-linked session related to that material (with a target of participants completing at least five sessions); and (vi) attrition (with a target of at least 80% of participants completing follow-up assessments)."

3. It would be helpful to know if the authors intend to establish what constitutes TAU for each participant, or establish a means of classifying interventions for a future trial (given wide variability in the type and availability of services for children with BI)?

The service use data that will be collected from participants' parents at baseline and at follow-up will be used to describe the profile of what constitutes TAU for participants in both groups. This is now clarified on page 15 of the revised manuscript: "In addition to informing the development of the economic evaluation methods, this will provide a profile of the resources and services that constitute TAU."

We are unlikely to have sufficient data from 25 participants to establish a comprehensive means of classifying the interventions that participants receive. However, this may be feasible in a future, larger trial, and we thank the reviewer for the suggestion.

4. 5 UK sites are mentioned. A little more detail regarding the nature of these sites would be helpful. In addition, are these sites the ones that will recruit to a full trial? Were they selected in order to be representative of the range of types of service which might be included in a future trial?

The sites were selected to be representative of the range of types of services which might be included in a future trial, with TAU ranging from no provision through to long-term out-patient multi-disciplinary teams. This has been summarised on page 5 in the manuscript: "These sites were selected to be representative of the range of types of services which might be included in a future trial, with TAU ranging from no provision through to services providing multidisciplinary out-patient care over the long-term."

5. It is noted that sites that fail to meet a recruitment target might not be included in a future trial. Given likely challenges with future recruitment this might not be the most helpful approach to take. Will the process measures allow for identification of barriers to recruitment and ways of improving recruitment in the future full trial, rather than simply disqualifying potential recruitment sites? In addition, no criteria is provided for 'poor recruitment'.

Each site was set a target of recruiting at least 5 participants. Table 1 has been amended to indicate this target (Item 3) and Item 8 has been rephrased to indicate the objective of identifying the barriers to recruitment and the support that might be required in a future definitive trial: "Do some sites need additional support to be able to recruit to target in a future trial?".

6. page 5 - how will the PI establish presence of EF difficulties?

Thank you. The PI will use clinical judgement regarding the likely presence of EF difficulties and infer from any direct and indirect assessments that they may have undertaken as part of their clinical work. We have included the following text in the revised manuscript (page 6): "The local PI is not required to undertake any EF assessment for the purposes of the study, but will use their clinical judgement, and infer from any assessments that they have undertaken, whether a participant is likely to be experiencing EF difficulties in their everyday lives."

7. page 5 - ICH needs writing out in full

We have deleted "ICH" from the manuscript (page 6) as upon further review, this was not relevant to the study methods.

8. top of page 6, Good Clinical Practice training might need an explanation for non-UK readers.

The following explanation has been included on page 6 of the revised manuscript: "GCP compliance ensures that the rights, safety and wellbeing of research participants are protected and that research data are reliable."

9. page 10, assessment of trial feasibility - as noted previously there is a lack of specific criteria against which feasibility might be assessed e.g. how will 'poor recruitment' be defined? Will measures of uncertainty of estimates of feasibility objectives (such as recruitment rate) be calculated i.e. 95% CI's? The authors state the sample size will be sufficient to establish feasibility - however it is not possible to establish if this is the case from the information provided, for example what sample size is required to estimate recruitment rate at an acceptable level of uncertainty?

Table 1, Item 3 has been changed to include the recruitment target for each site ("at least 5 participants"). The sample size of 25 will also allow us to estimate the standard deviations for continuous outcomes that may be used to inform the sample size for a definitive trial (Julious, 2005). The following text has been included on page 11 of the revised manuscript: "We will report the proportion of screened families who are found to be eligible, the proportion of eligible families who are recruited and randomised to their allocation, and the proportion of randomised participants who provide outcome data at follow-up, with 95% confidence intervals. Our sample size of 25 participants will allow us to estimate loss to follow-up (anticipated to be 20%) with a 95% CI of +/- 13 percentage points."

10. Table 3 - adult-completed secondary outcomes - can those that are child proxy measures be separately labelled to those which are measures of parent outcome?

Thank you. We have indicated, using an asterisk, the measures in Table 3, which refer to parent outcome. The following text has been included in the abbreviation list: "EQ5D-5L* EuroQol 5 Dimensions 5 Levels Questionnaire; PHQ9* Patient Health Questionnaire 9; GAD7* Generalised Anxiety Disorder 7-item." and "**Measures relating to parental outcome"

11. page 14 - the line 'The contribution of PPI members ...' - this is important, but seems somewhat vague, especially after the preceding paragraph setting out specific PPI contributions. Can the authors add specific plans regarding this valued PPI input in the study going forward?

We agree with the reviewer that PPI contribution is an important aspect of our feasibility study and we have further specified their planned involvement in the analysis, interpretation, and dissemination of the study findings. The following text has been included on page 16 of the revised manuscript: “Specifically, PPI members will be asked to contribute to data interpretation, assess whether the feasibility study has met its objectives, and support different approaches to dissemination of the study. They will also be asked to comment on potential changes for the main trial. The Director of Services and Innovation at CBIT will also support wider PPI in the development of the main trial.”

12. There is a lack of follow up time point - this is a significant weakness given the need for a follow up timepoint in a full trial, and also because longer term outcomes are especially important given the presence of brain injury within the developmental context. Some recognition of protocol limitations would be helpful in the discussion.

As with Reviewer 1, we agree with the reviewer 2, that a definitive trial would require longer follow-up than this feasibility study will provide. However, we think that the follow-up period proposed in this feasibility will allow us to receive valuable feedback from participants regarding the acceptability of outcome measures and their experiences of participating in the trial. The following text has been included on page 14 of the manuscript: “It is acknowledged that a definitive trial would require longer follow-up than this feasibility study will provide, but the feasibility trial will inform the acceptability of outcome measures and the experience of participating in the trial. The power calculation for a definitive trial with longer follow-up will account for a higher level of attrition than this feasibility trial.” We have also added the following study limitation on page 2: “This feasibility study was conducted over a short time period thus limiting the opportunity for a long-term follow-up. The definitive trial would include a follow-up period of at least 6 months.”

13. Online appendix - there is a bullet point without accompanying information

Thank you. This has been corrected.

VERSION 2 – REVIEW

REVIEWER	Fergus Gracey University of East Anglia, UK I have an honorary clinical position at one of the recruiting sites for this trial.
REVIEW RETURNED	10-May-2019

GENERAL COMMENTS	The authors have clearly given thorough consideration to the reviewers' comments and significantly improved the manuscript.
---